# Single-Cell Sequencing Reveals the Heterogeneity of Hepatic Natural Killer Cells and Identifies the Cytotoxic Natural Killer Subset in Schistosomiasis Mice

**DOI:** 10.3390/ijms26073211

**Published:** 2025-03-30

**Authors:** Fangfang Xu, Yuan Gao, Teng Li, Tingting Jiang, Xiaoying Wu, Zhihao Yu, Jing Zhang, Yuan Hu, Jianping Cao

**Affiliations:** 1National Institute of Parasitic Diseases, Chinese Center for Disease Control and Prevention, Chinese Center for Tropical Diseases Research, National Key Laboratory of Intelligent Tracking and Forecasting for Infectious Diseases, NHC Key Laboratory of Parasite and Vector Biology, WHO Collaborating Centre for Tropical Diseases, National Center for International Research on Tropical Diseases, Ministry of Science and Technology, Shanghai 200025, China; xufangfangx@163.com (F.X.); gyuan1028@126.com (Y.G.); drliteng@163.com (T.L.); jiangtingt16@163.com (T.J.); wuxiaoying2502@163.com (X.W.); yuzhihao0408@163.com (Z.Y.); zhangjing@nipd.chinacdc.cn (J.Z.); 2School of Global Health, Chinese Center for Tropical Diseases Research, Shanghai Jiao Tong University School of Medicine, Shanghai 200025, China

**Keywords:** single-cell RNA sequencing, NK cell, Thy1^+^NK, liver fibrosis, *Schistosoma japonicum*

## Abstract

*Schistosoma japonicum* eggs in the host liver form granuloma and liver fibrosis and then lead to portal hypertension and cirrhosis, seriously threatening human health. Natural killer (NK) cells can kill activated hepatic stellate cells (HSCs) against hepatic fibrosis. We used single-cell sequencing to screen hepatic NK cell subsets against schistosomiasis liver fibrosis. Hepatic NK cells were isolated from uninfected mice and mice infected for four and six weeks. The NK cells underwent single-cell sequencing. The markers’ expression in the NK subsets was detected through Reverse Transcription–Quantitative PCR (RT-qPCR). The proportion and granzyme B (Gzmb) expression of the total NK and Thy1^+^NK were detected. NK cells overexpressing Thy1 (Thy1-OE) were constructed, and functions were detected. The results revealed that the hepatic NK cells could be divided into mature, immature, regulatory-like, and memory-like NK cells and re-clustered into ten subsets. C3 (Cx3cr1^+^NK) and C4 (Thy1^+^NK) increased at week four post-infection, and other subsets decreased continuously. The successfully constructed Thy1-OE NK cells had significantly higher effector molecules and induced greater HSC apoptosis than the control NK cells. It revealed a pattern of hepatic NK cells in a mouse model of schistosomiasis. The Thy1^+^NK cells could be used as target cells against hepatic fibrosis.

## 1. Introduction

Schistosomiasis japonica is a significant zoonotic parasitic disease caused by *Schistosoma japonicum* infection. Approximately 250 million people worldwide are infected with schistosomiasis, and 280,000 people die of schistosomiasis every year [1,2]. Deposited eggs in the host’s liver induce egg granuloma formation, liver fibrosis, portal hypertension, and cirrhosis. Consequences include the loss of the labor force and a heavy economic burden on individuals and society [3,4]. The regulatory mechanism of schistosomiasis liver fibrosis has not been fully elucidated, resulting in a lack of effective treatment methods.

The mechanisms of schistosomiasis liver fibrosis are complex. The central event of liver fibrosis is hepatic stellate cell (HSC) activation induced by soluble egg antigens (SEAs) [5,6]. The activated HSC transforms into fibroblasts and synthesizes collagen fibers [7,8]. When collagen fiber synthesis exceeds its degradation, collagen is deposited in the liver, inducing liver fibrosis [9]. Liver fibrosis involves immune cells, proteins, microRNAs, and other factors [10]. Praziquantel is the only effective drug for schistosomiasis, but effective insecticidal treatment cannot wholly block the hepatic fibrosis process in schistosomiasis [11,12,13,14]. There is no effective treatment for schistosomiasis liver fibrosis, and related research is needed.

Senescent or early activated HSCs are eliminated by natural killer (NK) cells, a subset of innate immune cells that effectively fight fibrosis [15]. NK cells are abundant in hepatic non-parenchymal cells (NPCs). In mice, NK cells comprise approximately 10% to 30% of the lymphocytes in the liver, which is significantly higher than the proportion in other organs [16,17]. NK cells can kill activated HSCs or induce HSC apoptosis as the receptor of NK cells binds with ligands expressed on the surface of HSCs or through the secretion of interferon-gamma (IFN-γ) [18,19,20]. Recent data have indicated that special receptors are essential in the NK-cell-mediated killing of HSCs. Activation of specific receptors, such as metabotropic glutamate receptor 5 (mGluR5) and E-prostanoid 3 receptor (EP3), could augment NK cytotoxicity, promote NK cell killing of activated HSCs, and alleviate liver fibrosis [21,22]. Therefore, finding receptors that stimulate the NK cell killing function is an effective strategy to combat liver fibrosis.

NK cells exhibit immensely varied phenotypic and functional aspects. Single-cell sequencing is an effective method to distinguish NK cell subsets. Hepatic NK cells have been divided into multiple subsets in different disease models [23,24]. In liver fibrosis induced by *S. japonicum*, NK cells’ phenotypes, clusters, and functional changes are unclear. Thus, we established a mouse model of schistosomiasis, isolated NK cells from mouse liver, and performed single-cell RNA sequencing. Bioinformatic sequencing data analysis identified NK subsets essential for responding to *S. japonicum* infection. We screened the Thy1^+^NK subset, which exhibited a powerful killing function and could provide a new treatment approach against liver fibrosis induced by *S. japonicum* infection.

## 2. Results

### 2.1. Single-Cell RNA Sequencing Reveals the Heterogeneity of Hepatic NK Cells

A novel mouse model of *S. japonicum* infection was successfully constructed. Liver fibrosis in mice formed in the sixth week post-infection. Hepatic NPCs were isolated from uninfected and infected mice in the fourth and sixth weeks post-infection. NK cells were enriched from NPCs for single-cell RNA sequencing. The phenotypes, functions, and signaling pathways of the NK cell clusters in the liver were examined using bioinformatics. It depicts the single-cell sequencing strategy in Figure 1A. The number of cells in uninfected samples and at four and six weeks post-infection samples was 9191, 4860, and 6305, respectively. A total of 20,356 cells were sequenced effectively. After filtering out low-quality data, we acquired valid information on the transcriptome of approximately 18,436 cells. The ratio remaining after filtration was 90.57%. Single-cell sequencing data have been uploaded to the Genome Sequence Archive (https://ngdc.cncb.ac.cn/gsa (accessed on 31 December 2022)). The assigned accession of the submission is CRA013453.

Twenty-one clusters were created from 18,436 cells using the Uniform Manifold Approximation and Projection (UMAP) method. According to the expressed markers, these cells could be defined as NK cells (NK1.1^+^NKp46^+^), NKT cells (CD3^+^NK1.1^+^), neutrophils (Ly6G^+^CXCR2^+^), monocyte–macrophages (CD14^+^), and MDSCs (CD11b^+^Ly6G^+^) (Figure 1B). Four clusters, including C0, C6, C10, and C14, were defined as NK cells expressed with *Ncr1w* (*Nkp46*), *Klrk1* (*NKG2D*), and *Klrb1c* (*NK1.1*), but not *CD3*. The gene expression of C0 NK cells in different samples was compared. Klra7 had the highest gene expression level at the fourth week post-infection, and the expression level at the sixth week was lower than that of the fourth week post-infection (Figure 1C). The expression levels of other genes showed similar trends. The result suggested that hepatic NK cells were activated in the fourth week and inhibited in the sixth week after infection with *S. japonicum*.

### 2.2. Hepatic NK Cell Cluster and Functional Analyses

The number of NK cluster cells decreased dramatically with the progression of infection; however, the proportion of mature NK cells continued to increase (Appendix A). The gene expression, GO (Gene Ontology), and KEGG (Kyoto Encyclopedia of Genes and Genomes) analyses of each subpopulation were as follows.

Genes, such as *Klra4* (*Ly49d*), *Klra8* (*Ly49h*), and *Cma1*, were highly expressed in cluster 0 (Figure 2A). GO and KEGG enrichment analyses revealed leukocyte activation and NK-cell-mediated cytotoxicity (Figure 2B,C). The results indicated that cluster 0 may primarily exert lethal and aggressive effects, which could be defined as mature NK.

*Tnfrsf9*, *Ncr1*, and *Prf1* genes were highly expressed in cluster 6 (Figure 2A). GO enrichment analysis revealed the active functions of ribosomal RNA (rRNA) processing, rRNA metabolic process, and ribosome biogenesis (Figure 2B). The KEGG enrichment analysis demonstrated that the eukaryotic ribosome biogenesis pathway was active (Figure 2C). These findings suggested that this group of cells was in the protein synthesis stage and may be in the immature mitotic stage. We defined this cluster as immature NK.

*Klrc2*, *Txk*, and *Itga4* genes were highly expressed in cluster 10 (Figure 2A). The GO and KEGG enrichment studies determined that the nuclear lumen, nucleoplasm, and protein modification process had activated functions. NK-cell-mediated cytotoxicity was the most activated (Figure 2B,C). These results suggested that this group of cell metabolism of intracellular substances was active. We define this cluster as memory-like NK.

Many genes were highly expressed in cluster 14, including *Pf4*, *Ctla2a*, *CD9*, *Ppbp*, *Gng11*, *Nrgn*, *Tsc22d1*, *Klra9*, *Klra8*, and *Klra4* (Figure 2A). The GO and KEGG enrichment analysis revealed T cell activation, Th1 and Th2 cell differentiation, and modulation of cell adhesion pathways (Figure 2B,C). We define this cluster as regulatory-like NK.

Compared with the significance of pathway enrichment, the mammalian target of the rapamycin (mTOR) signaling pathway, NK-cell-mediated cytotoxicity, oxidative phosphorylation, the programmed death-ligand 1 (PD-L1) pathway, and platelet activation were the most active in memory-like NK cells. The most active signaling pathways in the regulatory-like NK cells were TNF, apoptosis, chemokine, FoxO, and transforming growth factor-beta (TGF-β) signaling (Figure 2D).

According to the gene expressions and enrichment analysis using the KEGG and GO databases, the hepatic NK cells could be categorized into four types, including mature NK (cluster 0), immature NK (cluster 6), memory-like NK (cluster 10), and regulatory-like NK (cluster 14) cells. 

### 2.3. Gene Set Variation Analysis (GSVA) of Signaling Pathways in NK Cells

The pathway expression levels in the four NK groups were compared based on the GSVA pathway scores. GSVA revealed characteristics of the various NK cells. Memory-like NK cells are a terminally differentiated NK cell type. The function of memory-like NK cells is complex. Compared to the other groups, memory-like NK cells had significantly increased inflammatory response, immunosuppressive response, cell migration, and complement activation. In contrast, proliferation and apoptosis of the memory-like NK cells were the weakest of the four NK groups (Figure 3B,E,F). Regulatory-like NK cells are negative feedback regulators. The immunosuppressive responses, such as TGFβ-smad4, DNA repair, mitochondrial biogenesis, and reactome apoptosis, were higher in regulatory-like NK cells than in memory-like NK cells (Figure 3F). Mature NK cells comprise the vast majority of hepatic NK cells. Complement cascade and inflammatory response levels in mature NK cells were higher than those in immature NK cells (Figure 3C). There were more ribosome, cell proliferation, apoptosis pathways, cell differentiation, and killing pathway activation in the immature NK cells compared to the other clusters (Figure 3A–C).

The GSVA analysis indicates that memory-like NK cells exhibited strong inflammatory responses and weak proliferative characteristics, regulatory-like NK cells highly expressed immune-inhibitory pathways, mature cells mainly activated reactome lectin pathway, complement activation, and the integrin pathway, and immature cells significantly activated proliferation and differentiation-related pathways.

### 2.4. NK Cell Re-Clustering and Functional Analyses

We reclassified the four NK group categories into 10 clusters (C0–C9; Appendix A) to identify the NK cell subsets associated with *S. japonicum* infection. The correspondence between the four groups of NK cells and the ten subsets is shown in Appendix A. The number and proportion of C3 and C4 increased significantly after the fourth week of *S. japonicum* infection. In comparison, the number and proportion of other clusters decreased significantly after infection (Appendix A). These findings suggest that C3 and C4 may be essential in resistance to *S. japonicum* infection. The highly expressed genes in C3 and C4 were analyzed (Figure 4A). The signal pathways and cell functions were enriched in C3 and C4 (Figure 4B,C). Appendix A shows the results of GO and KEGG enrichment analysis for the remaining eight clusters, except for C3 and C4.

Functional enrichment analysis of the 10 clusters of NK cells revealed that C0, C1, C2, C3, and C9 mainly involved mature NK cells. C0 controlled nitrogen compounds and the metabolic processes of nucleobases. Cytosolic large and small ribosomal subunits, the cytosolic ribosome, and the structural constituent of ribosome were active in cluster 1. C2 was related to the resistance against bacterial and fungal infection. Salmonella infection and bacterial invasion of epithelial cells were active. C3 was similar to CD8^+^ T cells, which played roles in antigen presentation with the MHC protein complex. C9 involved regulatory roles in transcription regulation, RNA metabolism, nucleic-acid-templated transcription, and others in intracellular macromolecule metabolism, biosynthesis, and inflammatory response. C5 and most of C4 comprised immature NK cells. The proteasome-related functions, such as proteasome core complex and threonine-type endopeptidase activity, were active in C4. C5 controlled ribosome synthesis, rRNA metabolism, and rRNA processing, among other processes. C7 belonged to regulatory-like NK cells, which are essential in fibrinogen binding, platelet aggregation, activation, and homotypic cell–cell adhesion functions. C6 and C8 comprised memory-like NK cells. The nucleoplasm, nuclear lumen, intracellular organelle lumen, and organelle lumen were all activated in C6 cells. Interferon-alpha (IFN-α) synthesis was activated in C8 cells.

Signal pathways from C0 to C9 were analyzed and compared (Figure 4D). In C5, pathways of the spliceosome and oxidative phosphorylation were activated. Proteasome, platelet activation, and apoptosis were activated in C4, C7, and C9, respectively. The FoxO signaling pathway, PD-L1 expression, and PD-1 checkpoint pathway in cancer and chemokine signaling pathway were activated in C6. The NF-κB signaling pathway was activated in C6 and C9.

Seven functional features were identified from the 79 signal pathways that were significantly activated (Figure 4E, Appendix A). They were followed by phagocytosis, metabolism, nucleic acid, structural-protein-related pathways, adverse regulatory signaling pathways, necrosis and apoptosis signaling pathways, inflammatory signaling pathways, and platelet-activation-related pathways. Among the 10 clusters, C2, C3, and C6 were the most complex. C2 and C3 cells were activated in five aspects, and C6 cells were activated in seven functions. The metabolic and inflammatory signaling pathways were specifically triggered in cluster 6.

In this part, the most interesting thing is that the proportion of C3 and C4 changes increased significantly with the progression of infection. They may play important roles in the schistosomiasis liver fibrosis. C3 is similar to CD8^+^ T cells involved in antigen presentation. The proteasome-related pathway in C4 has the most activity.

### 2.5. Analysis of Pseudo-Time in NK Clusters and Subsets

The cells in the infected samples were mainly distributed at the beginning and middle of the trajectory path (Figure 5A). C4 and C5 were at the start of the trajectory path, whereas clusters 6 and 8 were in a terminal state. Other clusters were in the middle of the trajectory path (Figure 5B,C). We evaluated the functions of NK cells on the toxicity, exhaustion, proliferation, and chemotaxis in the uninfected and infected samples at four and six weeks post-infection. The proliferation, toxicity, and chemotaxis indices of NK cells from the uninfected sample were higher than those from the sample six weeks post-infection. The exhaustion index did not vary among the three samples (Figure 5D).

Gene expression analysis was integrated with the trajectory analyses of the four NK groups. Memory-like NK cells were independent of the other three groups at the terminal stage of the trajectory pathway. Memory-like NK cells were the most mature cells and the most abundantly expressed of toxicity genes (*Ifng*, *Gzmc*, *Ncr1*, *Prf1*), chemotaxis genes (*Cxcr3*, *Cxcl10*, *Ccr5*, *Ccr1*), inhibition genes (*Tgfb1*, *Havcr2* (*Tim3*), *Tnfs10* (*Trail*), *Klrc1, Klra2, Fas1*, *Tigit*), proliferation genes (*Cd69*, *mTOR*), activated receptors (*Cd226*, *Klra2* (*Ly49b*), *Ncr1, Klrk1* (*NKG2D*), *Klrc2* (*NKG2C*)), and platelet-activation-related factors (*Pf4*, *Ppbp*, *Gng11*, *Nrgn*, *Itga2b*) (Figure 5E, Appendix A).

Immature NK cells were detected at the beginning of NK cell development, consistent with the expression of some genes. They displayed inhibited receptor genes (*Ctla4, Tigit, Klra1* (*Ly49a*), *Il-10*, *Tgfb1*, *Lamp1*), chemotaxis genes (*Cxcr3*, *Cxcl10*, *Ccl5*, *Ccr5*), toxicity genes (*Eomes*, *Xcl1*, *Gzma*, *Gzmm*, *Gzmb*, *Cd160*), and the proliferation gene Il21r (Figure 5E, Appendix A).

Mature NK cell development occurred later than that of immature NK cells in the middle of the trajectory pathway. It was consistent with the expression of many genes, such as inflammatory genes (*S100a4*, *S100a6*, and *Irf8*), active receptors (*Klra3, Cma1, Klrg1, Klra4* (*ly49d*), *Klrg1*, *Prf1*), killing associated transcription factors (*Klf2, Tbx21*), and chemotaxis (*Ccl5*, *Cx3cr1*) (Figure 5E, Appendix A).

The development of regulatory-like NK cells occurred in the middle of the trajectory pathway. The development coincided with the periods of immature and mature NK cells. Development of regulatory-like NK cells was consistent with the periods of many genes, including those for activator receptors (*Klrk1* (*NKG2D*), *Klrc2* (*NKG2C*), *Klra4* (*Ly49d*)), inhibitory receptors (*Klrd1* (*CD94*), *Klra3* (*Ly49c*)), toxicity genes (*Eeomes*, *Zeb2*, *Prf1*, *Klf2*), and chemotaxis genes (*Cxcr4*, *Ccl5*) (Figure 5E, Appendix A).

The expression of *zeb2* was consistent with the periods of differentiated NK cells, such as mature NK, memory-like NK, and regulatory-like NK cells. The periods of expression of *klrg1*, *xcl1*, and *klrc1* were the same during the separate development of mature NK, immature NK, and memory-like NK cells. *Ccl5* expression coincided with the development of regulatory-like NK cells, immature NK cells, and mature NK cells (Appendix A).

The pseudo-time analysis reveals that persistent infection suppresses NK function. The immature NK cells are at the starting point of differentiation. The mature NK cells and regulatory-like NK cells are located in the middle of the differentiation trajectory. The memory-like NK cells are terminally differentiated NK, which have a complex function.

### 2.6. The Expression of Specific Markers of Each NK Cluster

We identified ten NK subgroup-specific marker molecules (*Kcnj8, Cmah, Ccnd2, Cx3cr1, Thy1, Tnfrsf9, Zbtb20, Gng11, CD226, Egr3*) from C0 to C9 and assessed their expression changes during infection using RT-qPCR. The results revealed a significant decrease in the expression levels of eight gene markers of C0, C1, C2, C5, C6, C7, C8, and C9 post-infection. At the same time, *Cx3cr1* (C3 marker) expression continued to exhibit an upward trend at weeks four and six post-infection. *Thy1*(C4 marker) expression increased significantly at week four and decreased at week six post-infection (Figure 6). The expression of marker genes aligned with the single-cell sequencing results of the NK subpopulation proportions.

### 2.7. The Changes in Primary NK Cells and C4 After S. japonicum Infection

The NK cell population within hepatic lymphocytes was defined using CD3^-^NK1.1^+^ (Figure 7A). The flow cytometry results revealed a sustained increase in the hepatic NK ratio at weeks four and six post-infection (Figure 7B). Gzmb expression in the NK cells increased four weeks post-infection and decreased six weeks post-infection. The result indicated that the NK cell cytotoxicity changed at different times (Figure 7C). The expression level of thy1 in the C4 subset is the highest, and it can be regarded as the surface marker of the C4 cluster. Compared to the uninfected group, the proportion of C4 (Thy1^+^NK) cells within the total NK cells increased at week four post-infection and decreased at week six post-infection. The altered Thy1 expression detected through flow cytometry was consistent with that of the RT-qPCR and single-cell sequencing (Figure 7D). At week four post-infection, the changes in Gzmb secretion in C4 cells were consistent with that of the total NK cells (Figure 7E). The result indicated that Thy1^+^ NK cell cytotoxicity increased four weeks post-infection but was attenuated six weeks post-infection.

### 2.8. Thy1^+^ NK Cell Cytotoxicity Function Assay In Vitro

The role of Thy1^+^NK was explored by constructing NK92 cells that overexpressed Thy1. Figure 8A shows that both groups of cells exhibit green fluorescence, indicating that the lentivirus was successfully transfected into the cells. Furthermore, the RT-qPCR analysis confirmed the successful overexpression of Thy1 (Thy1-OE) in the NK92 cells (Figure 8B). Flow cytometry revealed that interleukin-12 (IL-12), interleukin-15 (IL-15), and interleukin-18 (IL-18) induced NK cell activation and significantly elevated the Gzmb and Perforin (Prf1) expression levels in the Thy1-OE NK cells compared to the control NK cells at 24 h and 48 h. The RT-qPCR results were consistent with the flow cytometry results (Figure 8C–J). Following 48 h of stimulation by IL-12, IL-15, and IL-18, Gzmb and Prf1 expression in the NK92 and Thy1-OE NK cells increased significantly compared to that following 24 h of stimulation. Furthermore, the NK cells were stimulated for 24 h with SEA, and the Gzmb and Prf1 expression levels in the Thy1-OE NK cells increased significantly compared to the control group (Figure 8K–N). These results suggested that the Thy1-OE NK cells had more potent cytotoxic activity than the control NK cells.

The NK cells were co-cultured for 6 h with LX-2 cells. The early and late apoptosis rates of the LX-2 cells co-cultured with Thy1-OE NK cells were more significant than those co-cultured with NK cells (Figure 9A). Furthermore, Western blotting revealed significant down-regulation of the anti-apoptotic protein Bcl2 and notable up-regulation of the pro-apoptotic protein Bax. The Bcl2: Bax ratio decreased significantly in the LX-2 cells co-cultured with Thy1-OE NK cells compared to that of the LX-2 cells co-cultured with NK cells (Figure 9B,C). These results indicated that the Thy1-OE NK cells exhibited a significantly enhanced capacity to induce LX-2 cell apoptosis compared to NK cells.

## 3. Discussion

The development of liver fibrosis is regulated by various factors, including immune cells, cytokines, etc. [25]. M2 macrophages induce hepatic stellate cells to undergo autophagy through the PGE2/EP4 pathway to promote liver fibrosis in nonalcoholic fatty liver disease mice [26]. Macrophages are induced to undergo M1-type polarization by MyD88 in hepatic stellate cells, which can enhance liver fibrosis [27]. CD8^+^ tissue-resident memory T cells inhibit hepatic fibrosis by inducing hepatic stellate cell apoptosis [28]. Natural killer cells (NK) are important immune cells in the liver. In recent years, the anti-fibrotic effect of NK cells has attracted much attention from researchers.

NK cells are abundant in NPCs, accounting for 10% of liver lymphocytes in mice and 30% to 50% in humans and rats [23]. NK cells kill early activated HSCs by secreting IFN-γ or inducing HSC apoptosis through Fas-FasL, which halts the process of liver fibrosis [17,19]. Significant inhibition of NK function has been described in patients with liver fibrosis and mouse models of liver fibrosis [29,30]. Decreased secretion of IFN-γ, TNF-α, granzyme, and perforin and inhibited killing of HSCs by NK cells has been demonstrated [31,32,33,34]. The expression of inhibitory receptors, such as Tigit, Lag3, and TIM3, increases significantly [30,35]. IFN-γ, perforin, and granzyme expression increased four weeks post-infection and decreased substantially six weeks post-infection [29]. Gene expression in NK cells increased four weeks post-infection and decreased six weeks post-infection. The findings suggest the activation and inhibition of hepatic NK cells after four and six weeks of being infected with *S. japonicum*, respectively. This trend aligned with our earlier findings on NK cell function [29].

Increasing data indicate that intrahepatic NK cells are a heterogeneous population [36]. Wijaya demonstrated that KLRG1^+^ NK cells relied on the bone-bridging proteins TRAIL and CD44 to promote HSC apoptosis and mitigate hepatic fibrosis [19]. Tao demonstrated that EP3 augmented NK cell adhesion and cytotoxicity to HSC in a murine model of liver fibrosis [22]. Previously, we reported that knocking down TIGIT enhanced NK cell function and attenuated liver fibrosis in schistosomiasis [37]. Therefore, identifying potent killing clusters within intrahepatic NK cells represents an effective strategy against liver fibrosis.

In the present study, a mouse model of schistosomiasis was established. Hepatic NK cells were enriched from NPCs. Single-cell RNA sequencing was performed. According to marker expression, GO, and KEGG analyses, four NK groups were identified: mature NK, immature NK, memory-like NK, and regulatory-like NK cells.

Cluster 0 was defined as mature NK. Genes, such as *Klra4* (*Ly49D*)*, Klra8* (*Ly49H*), and α-chymase (*CMA1*), were specifically expressed and could be considered markers of this cell population. Ly49D and Ly49H are killer cell lectin-like receptors of NK cells [38]. Ly49D recognizes MHC-I molecules and promotes Ly49H^+^ cell secretion of IFN-γ [39]. CMA1 is a serine protease belonging to the peptidase family S1. CMA1 secreted by uterine NKs is pivotal to the vascular changes required to support pregnancy [40]. GO and KEGG enrichment analyses revealed that mature NK may primarily exert lethal and aggressive effects. GSVA scores indicated that the functions of mature NK cells mediating inflammation and cell migration were more robust, while the killing effects and differentiation were weaker than those of immature NK cells. The analysis of pseudo-time showed that this cluster was present in the middle of the trajectory pathway, with high expressions of many activated receptors and inflammation-associated cytokines.

Cluster 6 was defined as immature NK. Genes, such as *Prf1* and *Tnfrsf9* (*CD137*), were specifically expressed and could be considered markers of this cell population. Prf1 is a cytolytic protein secreted by NK cells, and its secretion level is closely related to the cytotoxicity of NK cells [41]. Activation of CD137 reportedly counteracts TGF-β inhibition of NK cell antitumor function and induces reprogramming of cellular mitochondrial morphology and function [42,43]. The GO enrichment analysis in the present study indicated that the function of the ribosome was active. This finding suggests that immature NK cells are in the mitotic stage. Analysis of pseudo-time showed that this cluster (re-cluster 4 and 5) was mainly distributed at the beginning and middle of the trajectory path. This indicates that this cell population is in the early stages of differentiation.

Cluster 10 was defined as memory-like NK. Genes, such as *Klrc2, Txk*, and *Itga4*, could be considered markers of this cell population. Klrc2 (NKG2C) is a marker of memory NK cells [44]. Txk, a Tec family of tyrosine kinases member, translocates from the cytoplasm into nuclei after activation. It precisely regulates IFN-γ transcription [45]. Integrin alpha 4 (Itga4) acts in cell–cell adhesion via plasma membrane cell adhesion molecules [46]. In the present study, GO and KEGG analyses revealed activation of protein modification and killing functions in memory-like NK cells. The rapid accumulation and long-term maintenance of memory-like NK cells may effectively treat virus infections [47]. The pseudo-time analysis suggests that memory-like NK cells are terminally differentiated cells.

Cluster 14 was defined as regulatory-like NK. Genes, such as *Pf4, Ctla2a*, and *CD9*, could be considered markers of this cell population. Non-platelet-derived Pf4 (CXCL4) promotes Treg proliferation and down-regulates PD-1 expression in Tregs [48]. Whether Pf4 has a similar effect on NK cells is unclear. Cytotoxic T lymphocyte antigen 2 (CTLA2a) is a tolerogenic mediator of TGF-β-mediated iTreg induction [49]. CD9 displays integrin activity and negatively regulates NK cytotoxicity and cytokine secretion [50]. CD9 has been identified as a marker of Breg secretion IL-10 [51]. Pathways of FoxO signaling, TGF-β signaling, and apoptosis were active in regulatory-like NK cells. This indicates that this cluster mainly plays the role of a negative regulatory function.

The four NK cell groups were re-clustered and divided into ten subsets (C0–C9) to identify NK subsets related to *S. japonicum* infection. Only C3 and C4 numbers increased; other clusters decreased significantly at four weeks post-infection. The findings implicated C3 and C4 as essential to resistance to *S. japonicum* infection. The expression of ten marker gene clusters (*Kcnj8, Cmah, Ccnd2, Cx3cr1, Thy1, Tnfrsf9, Zbtb20, Gng11, CD226, Egr3*) was validated using RT-qPCR, and the observed expression patterns were consistent with the sequencing results.

Single-cell sequencing successfully characterized the C4 of immature NK cells, representing an NK cell subset exhibiting robust cytotoxic activity. Sequencing analysis revealed that C4 had high expression of the *Thy1, CD7, Serpina3g, CD27*, and *Xcl1* genes. GO analysis revealed the activation of the proteasome function of C4. This result suggested that the killing function of C4 NK cells involved the proteasome.

Thy1 is a glycoprotein anchored to cell membranes by GPI molecules and contains RGD-like motifs (RLDs) that bind to integrins [52,53,54]. Thy1 is a crucial regulator of cell–cell and cell–matrix interactions and is critical in neural regeneration, cell adhesion and migration, and the inflammatory and fibrotic processes [55,56,57,58]. Itami found that Thy1 may bind to integrin β3 [59]. Thy1 inhibition blocked the adhesion of tumor cells to mouse lymphatic endothelial cells [60]. In our study, Thy1-overexpressed NK cells have a strong ability to kill LX-2. It is speculated that thy1 might enhance the adhesion of NK cells to LX-2, thereby killing more LX-2 cells. Kupz observed that Thy1^+^NK cells could secret protective IFN-γ in the early stages of Salmonella infection, enhancing the host’s antimicrobial immunity [61]. In this study, we used SEA to stimulate NK cells in vitro to mimic the stimulation of NK received after infection in vivo. After stimulation by SEA, Thy1-overexpressed NK cells secreted more Gzmb and Prf1. This indicates that Thy1 can enhance the cytotoxicity of NK cells, but the specific mechanism involved remains unclear and requires further study.

In our study, we observed that the proportion of C4 NK cells (Thy1^+^NK) increased significantly in the hepatic NK cells at four weeks post-infection and decreased at six weeks post-infection. The cytotoxic function of the Thy1^+^NK cells was active at four weeks post-infection and was inhibited at six weeks post-infection. The experiments in vitro demonstrated that Thy1 overexpressed could enhance cytotoxic activity. However, the mechanism through which Thy1 enhances the cytotoxicity of NK cells is unclear. This study provides clues to further search for NK cell clusters with high cytotoxicity. However, the limitation is that we do not explore Thy1^+^NK’s role in liver fibrosis in vivo. In the future, we will continue to verify Thy1^+^NK’s role against liver fibrosis in animals.

## 4. Materials and Methods

### 4.1. Ethics Statement

All animal experiments were performed in strict accordance with the regulations for the administration of affairs concerning experimental animals in China, and efforts were made to minimize suffering. All procedures performed on animals in this study were approved by the Laboratory Animal Welfare and Ethics Committee of the National Institute of Parasitic Diseases, Chinese Center for Disease Control and Prevention (Chinese Center for Tropical Diseases Research) (approval ID: IPD 2020–10).

### 4.2. Mouse Models of Infection

Six-week-old C57BL/6 mice were provided by the Shanghai Slack Laboratory Animal Co., Ltd. (Shanghai, China). Mice were housed under specific pathogen-free conditions in the animal room of the National Institute of Parasitic Diseases, Chinese Center for Disease Control and Prevention. Cercariae of *S. japonicum* (Chinese mainland strain) was provided by the snail room of our institute. Eighteen C57BL/6 mice were randomly divided into three groups examined before infection and four and six weeks after infection. Infection was performed using an established technique [24]. Each infected mouse received 20 ± 1 cercariae via abdominal skin. Each sample was a mixture of hepatic NK cells from six mice in each group.

### 4.3. Enrichment of NK Cells

C57BL/6 mice were sacrificed via cervical dislocation after anesthesia. After removing the remaining blood, the liver was removed, minced with scissors, and dissociated into single-cell suspensions using an Ultra Turrax tube disperser (IKA, Königswinter, Germany). The suspension was centrifuged to collect cells. Cells were separated through differential gradient centrifugation and washed twice with 1× Dulbecco’s Phosphate-Buffered Saline(Gibco, Grand Island, NY, USA). Red blood cells were lysed using BD Pharm Lyse Lysing Buffer (BD Biosciences, San Jose, CA, USA) to obtain NPCs.

The concentration of NPCs was adjusted to 1 × 10^7^ cells. NK cells were enriched using a cell isolation kit (Miltenyi Biotec, Bergisch Gladbach, Germany). A brief description follows. After centrifuging, the cell pellet was resuspended in 40 µL of buffer per 10⁷ total cells. Ten microliters of NK cell biotin–antibody cocktail were added and incubated for 5 min in the refrigerator. After washing, 80 µL of buffer and 20 µL of anti-biotin microbeads were incubated for 10 min. The cell suspension was applied to the column. The eluant containing unlabeled cells, which were enriched NK cells, was collected. Dead cells were removed from the enriched NK cells using a dead cell removal kit (Miltenyi Biotec, Bergisch Gladbach, Germany).

### 4.4. Single-Cell RNA Sequencing and Analysis

According to the manufacturer’s protocol, freshly isolated cells were analyzed using a chromium single-cell 3′v3 library chip (10× Genomics, Pleasanton, CA, USA). cDNA sequencing libraries were prepared according to cell lysis, reverse transcription, cDNA amplification, product extraction, fragmentation, and PCR and then cyclized into single-strand DNA libraries. Sequencing was performed on a Novaseq 6000 apparatus (Illumina, San Diego, CA, USA) at the IZKF genomics facility of the RWTH Aachen University in Aachen, Germany. Sequencing used 2 × 150 bp paired-end reads. In the FASTQ format, raw sequence reads from samples were processed and aligned to the mouse reference transcriptome (https://www.10xgenomics.com/ (accessed on 31 December 2022)) using the Cell Ranger v 4.0.0 pipeline with default parameters. Gene expression matrices were merged using Seurat package v3 to remove empty droplets, probable doublets, and low-quality cells. Expression values were normalized for total UMIs per cell, and counts were log-transformed using the Seurat Normalize Data function to account for differences in sequencing depth across samples.

### 4.5. Clustering and Identification of Markers

Seurat v3 was used to generate the count expression matrix. The suitable principal components were selected for clustering from 30 principal components. Clustering was performed using the FindClusters function on the K-nearest neighbor graph model. We used the FindMarkers function in the Seurat package to detect cluster-specific expressed genes with default parameters. Known marker genes were used to identify cells. We reran the Seurat cluster workflow for each cell type to identify sub-cell types.

### 4.6. Function Enrichment Analysis

Differentially expressed genes were selected using the FindMarker function in the Seurat package of different samples. GO enrichment analysis for these genes was performed using the TopGO package in R. KEGG information was downloaded from the KEGG website. The enricher function in the ClusterProfiler package was used to analyze KEGG enrichment. Pathway analyses were performed to investigate the biological states of different clusters on the 50 hallmark pathways described in the molecular signature database.

The AddModuleScore function in Seurat was used to calculate module scores for the gene expression program in a single cell. The analyzed genes were binned, and the control gene was randomly selected from each bin based on the average expression. The average expression value was determined based on the gene set at the single cell level of expression minus the aggregated expression of the control gene set. Gene sets were obtained from the MSigDB database (https://www.gsea-msigdb.org/gsea/msigdb/ (accessed on 31 December 2022)).

### 4.7. Pseudo-Time Analyses

The NK cell developmental trajectory was reconstructed using the R package Monocle2 v2.99.3. The “Relative2abs” function in Monocle2 converts TPM into normalized mRNA counts. The UMI matrix was used as input, and Seurat was used to detect variable genes obtained from NK cell types to sort cells in pseudo-time. After the cell trajectories were constructed, differentially expressed genes along the pseudo-time were detected using the DifferentialGeneTest function.

### 4.8. Construction of Thy1-Overexpressing NK92 Cells

The lentiviral vector pLent-EFla-FH-CMV-copGFP-P2A-Puro was selected to package the Thy1 plasmid. EFla activates Thy1 expression, and Thy1 is nonfused with GFP. Lentivirus overexpressing the human *THY1* gene and empty plasmid control lentivirus were packaged and synthesized by Weizhen Biological Company (Jinan, China). NK92 cells (human cell line) from Pricella (Wuhan, China) were infected with the target and control viruses at a multiplicity of infection (MOI) of 100. After 72 h of infection, puromycin was added to a final concentration of 4 μg/mL to screen for positive cells. The negative control (NC-NK92) and Thy1-overexpressing NK92 cell lines (Thy1-OE NK92) were obtained after two weeks of continuous screening, where the medium was replaced every 48 h.

### 4.9. Detection of NK Cells’ Function

The NC-NK92 and Thy1-OE NK92 cells were stimulated with triple cytokines of IL-12 (50 ng/mL), IL-15 (50 ng/mL), and IL-18 (100 ng/mL) (MedChemExpress, Shanghai, China) for 24 h and 48 h. The expression changes of Gzmb and Prf1 in the cells were assessed using flow cytometry and RT-qPCR. The NK cells were stimulated with SEA (120 μg/mL) for 24 h. The expressions of Gzmb and Prf1 were detected through flow cytometry and qPCR. The LX-2 (human cell line) was purchased from the Shanghai Cell Bank of the Chinese Academy of Sciences and cultured in complete Dulbecco’s modified Eagle’s medium (DMEM) containing 10% fetal bovine serum (Gibco, Grand Island, NY, USA). The NK92 cells were co-cultured with the LX-2 cells at a ratio of effector cells to target cells of 5:1 for six hours. The co-cultured LX-2 cells were collected, and an apoptosis assay kit (Absin, Shanghai, China) was used to measure the apoptosis rate according to the manufacturer’s instructions. The expression of apoptosis-related proteins was detected through Western blotting.

### 4.10. Reverse Transcription–Quantitative PCR (RT-qPCR)

Primary NK cells were isolated from the liver of mice, followed by total RNA extraction and reverse transcription to complementary DNA (cDNA). Subsequently, the expression of marker genes in the cluster (C)0 to C9 NK cell subpopulations was assessed using RT-qPCR (Appendix A). Gzmb and Prf1 expressions in the Thy1-OE NK92 and NK92 control stimulated with SEA or triple IL-12, IL-15, and IL-18 cytokines were also assessed using RT-qPCR (Appendix A).

### 4.11. Flow Cytometry

The cells were stained by adding a diluted solution of FVS-BV605 (1:1000 dilution) (BD Biosciences, San Diego, CA, USA) to the cell suspension. The mixture was incubated at room temperature and protected from light for 15 min. After washing, the cells were incubated with diluted surface antibodies of CD3-APC-cy7, NK1.1-FITC, and Thy1-BV421 (1:100 dilution) (BD Biosciences, San Diego, CA, USA) for 30 min at room temperature in a light-free environment. After fixing, the cells were stained with diluted intracellular antibodies Gzmb-PE-cy7, Gzmb-PE, and Prf1-PE-cy7 (Thermo Fisher Scientific, Waltham, MA, USA) at 4 °C for 45 min away from light. The cells were suspended and analyzed using a flow cytometer. LX-2 cells were collected through centrifugation at 2000 rpm for 5 min. After resuspension, cells were incubated separately with antibodies including annexin V-APC and PI-7-AAD (Absin, Shanghai, China). Finally, LX-2 cell apoptosis was detected using flow cytometry.

### 4.12. Western Blotting

The LX-2 cells were harvested and lysed. The protein concentration was determined using the bicinchoninic acid (BCA) method (Beyotime, Shanghai, China). The protein samples underwent sodium dodecyl sulfate–polyacrylamide gel electrophoresis (SDS-PAGE) and were transferred to a PVDF membrane. After being blocked for 1 h, the samples were incubated with primary antibodies, including anti-GAPDH (CST, Danvers, MA, USA), anti-Bcl2 (CST, Danvers, MA, USA), and anti-Bax (ProteinTech, Wuhan, China) at 4 °C overnight. Subsequently, the samples were incubated with the secondary antibody, a horseradish peroxidase (HRP)-conjugated anti-rabbit IgG antibody (Starter, Hangzhou, China), at room temperature for 1 h. The proteins were visualized using an electrochemiluminescence (ECL) chemiluminescent substrate and analyzed semi-quantitatively using ImageJ software v1.54.

### 4.13. Statistical Analysis

All data are expressed as the mean ± standard deviation. The two groups were compared using the *t*-test. Statistical analysis was performed using GraphPad Prism 9.5, and flow data were analyzed using FlowJo v10.8.

## 5. Conclusions

In the present study, the profile of hepatic NK cells was elucidated in the mice infected with *S. japonicum*. We determined that the hepatic NK cells could be divided into immature NK, mature NK, memory-like NK, and regulatory-like NK cells. These four groups were further re-clustered as cluster 0 to 9 subsets. Clusters 3 and 4 increased four weeks after *S. japonicum* infection, which may indicate the existence of special NK cells resistant to *S. japonicum*. In vitro, the NK cells that overexpressed Thy1 exhibited potent cytotoxicity. This investigation provides a solid basis for further exploration of the role of NK cells in schistosomiasis-induced liver fibrosis.

## Figures and Tables

**Figure 1 ijms-26-03211-f001:**
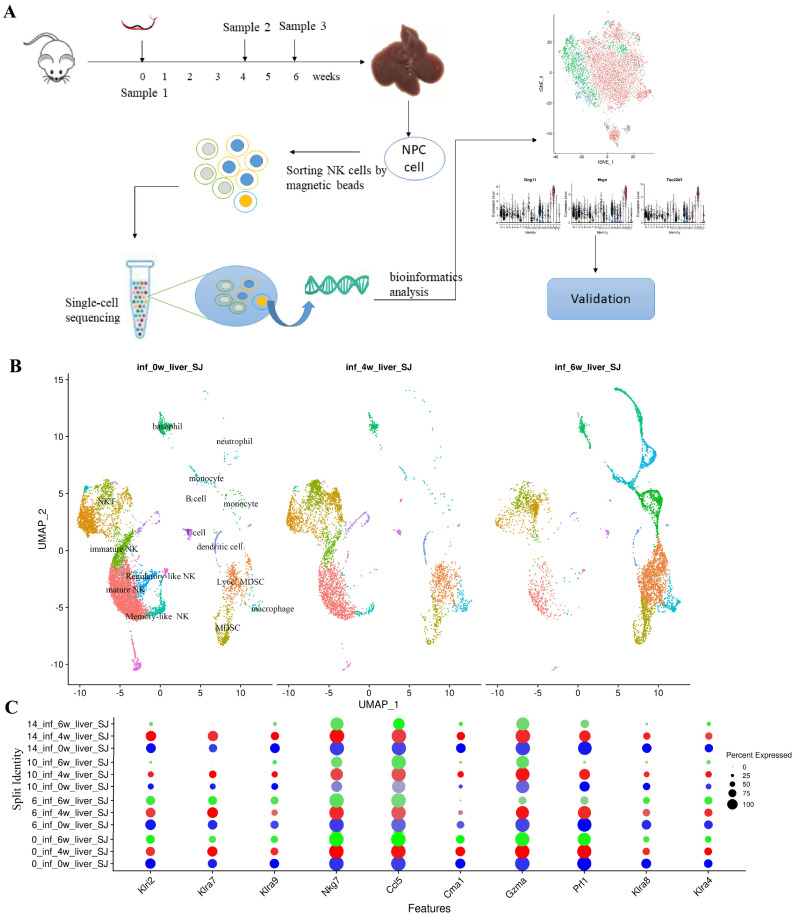
Single-cell RNA sequencing profile of enriched hepatic NK cells in mice infected with *S. japonicum*. (**A**) Schematic representation of the experimental design. (**B**) Visualization of single-cell data using UMAP, showing the annotation and color codes for cell types in enriched hepatic NK cells. (**C**) Gene point map of the top 10 markers of 4 clusters in three samples. inf_0w_liver_SJ, inf_4w_liver_SJ, and inf_6w_liver_SJ” represent uninfected and four- and six-week post-infection samples, respectively, and the numbers (0, 6, 10, 14) in front of them represent the names of NK clusters. The blue, red, and green dots represent samples that were uninfected and four- and six-week post-infection samples, respectively, and the size of the dots represents the level of gene expression.

**Figure 2 ijms-26-03211-f002:**
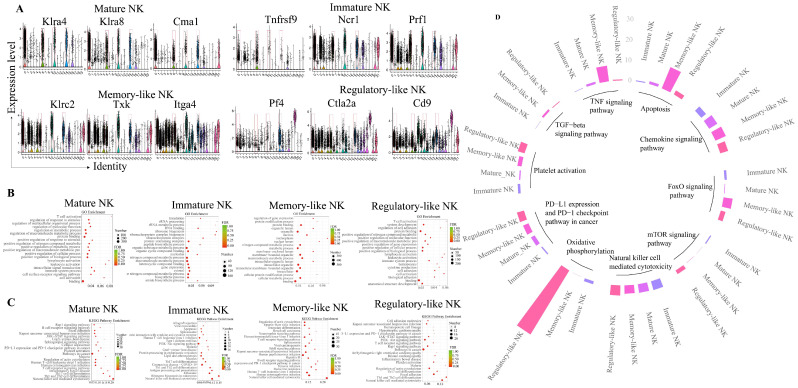
Identification of four clusters of NK cells in the liver of mice through single-cell RNA sequencing. (**A**) Violin plots of three markers with high expression level in each of the four NK cell clusters. Four clusters of NK are circled in red boxes in each violin plots, from left to right, cluster 0, cluster 6, cluster 10, and cluster 14. (**B**) GO enrichment analysis of the four NK cell clusters. (**C**) KEGG enrichment analysis of the four NK cell clusters. The size of the points represents the number of differentially expressed genes of the term; the color of the points represents the value of FDR. (**D**) Comparison of significant pathways among the four NK clusters. Different colors of the bar graph represent the four clusters of NK: Mature NK, Immature NK, Memory-like NK, and Regulatory-like NK. The height of the bar graph represents the pathway expression level.

**Figure 3 ijms-26-03211-f003:**
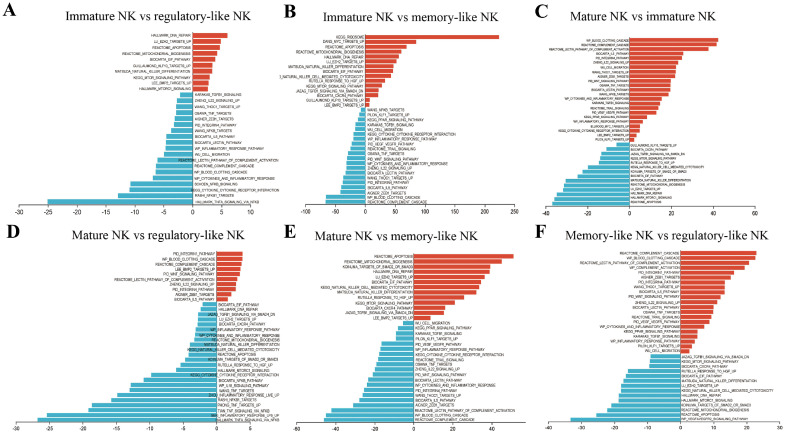
GSVA scores reveal differences in signaling pathway expression among the four NK clusters. (**A**) Comparison of signaling pathway expression between immature and regulatory-like NK cells. (**B**) Comparison of signaling pathway expression between immature and memory-like NK cells. (**C**) Comparison of signaling pathway expression between mature NK and immature NK cells. (**D**) Comparison of signaling pathway expression between mature NK and regulatory-like NK cells. (**E**) Comparison of signaling pathway expression between mature NK and memory-like NK cells. (**F**) Comparison of signaling pathway expression between memory-like NK and regulatory-like NK cells. The red indicates that the pathway expression level is higher in this cluster, while the blue indicates that the pathway expression level is lower in this cluster.

**Figure 4 ijms-26-03211-f004:**
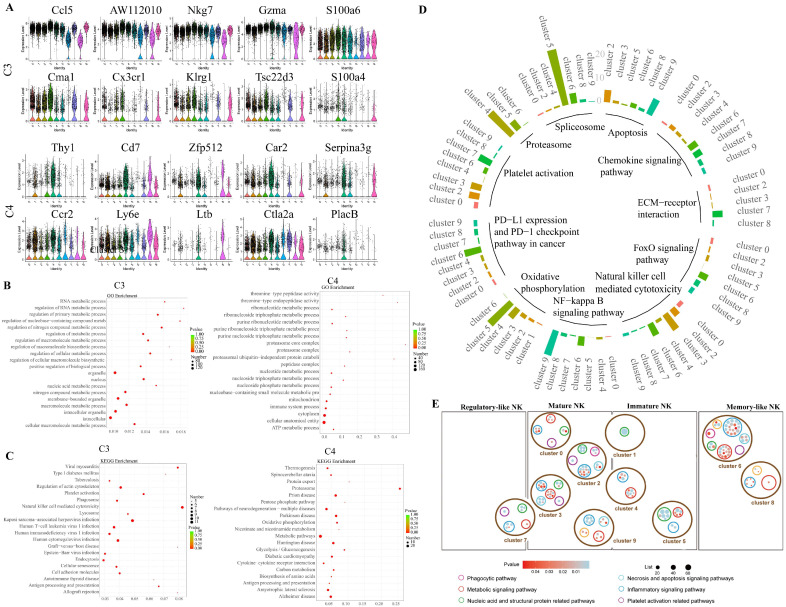
Re-clustering and functional analysis of hepatic NK cells. (**A**) Violin plots of the top ten markers in cluster 3 and cluster 4. (**B**) GO enrichment analysis of cluster 3 and cluster 4. (**C**) KEGG enrichment analysis of cluster 3 and cluster 4. The size of the points represents the number of differentially expressed of genes of the term; the color of the points represents the value of FDR. (**D**) Comparison of significant pathways among the ten subsets. Different colors of the bar graph represent the ten clusters of NK: cluster 0 to cluster 9. The height of the bar graph represents the pathway expression level. (**E**) Analysis of six functions correlated with signal pathways among the four clusters and ten subsets of NK cells. The size of the circle represents the number of signal pathways. The color of the circle corresponds to different cellular functions.

**Figure 5 ijms-26-03211-f005:**
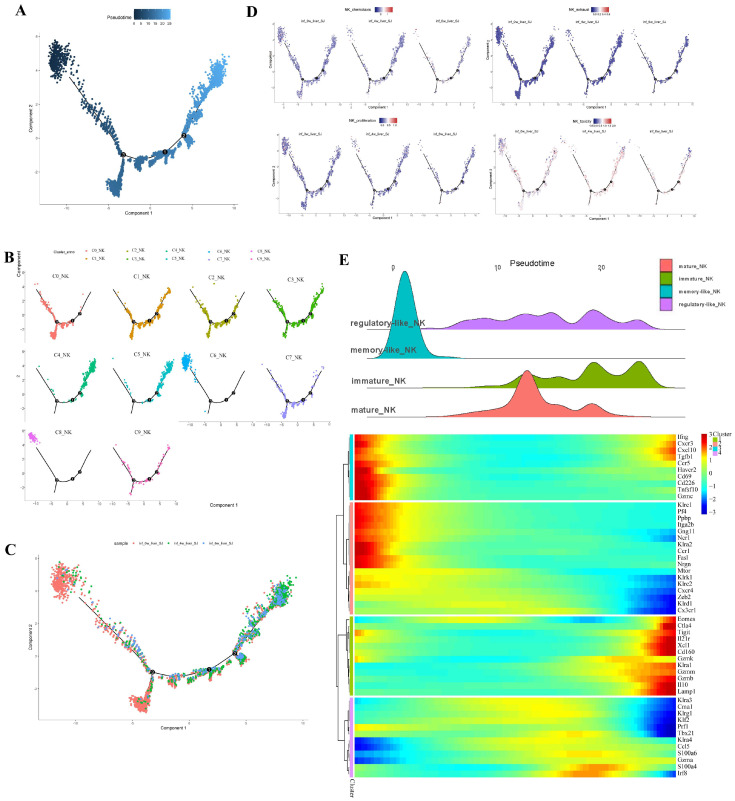
Pseudo-time analysis of NK clusters and subsets. (**A**) Monocle pseudo-time of hepatic NK cells. The numbers 1, 2, and 3 in the circle represent branch points 1, 2, and 3, respectively. (**B**) Monocle pseudo-time of different NK cell clusters. (**C**) Monocle pseudo-time of different samples. (**D**) Monocle pseudo-time indices of toxicity, exhaustion, proliferation, and chemotaxis. (**E**) Trajectory analysis of the four NK cell groups combined with gene expression.

**Figure 6 ijms-26-03211-f006:**
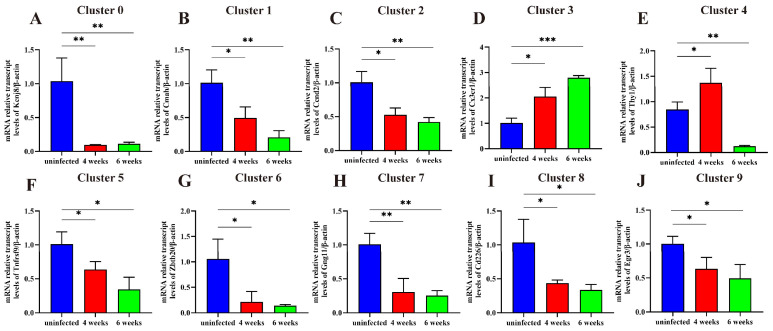
Changes in the expression levels of C0 to C9 marker genes in hepatic NK cell subsets were observed in uninfected mice and mice at four and six weeks post-infection. (**A**) Changes in *Kcnj8* expression (C0) in hepatic NK cells. (**B**) Changes in *Cmah* expression (C1) in hepatic NK cells. (**C**) Changes in *Ccnd2* expression (C2) in hepatic NK cells. (**D**) Changes in *Cx3cr1* expression (C3) in hepatic NK cells. (**E**) Changes in *Thy1* expression (C4) in hepatic NK cells. (**F**) Changes in *Tnfrsf9* expression (C5) in hepatic NK cells. (**G**) Changes in *Zbtb20* expression (C6) in hepatic NK cells. (**H**) Changes in *Gng11* expression (C7) in hepatic NK cells. (**I**) Changes in *Cd226* expression (C8) in hepatic NK cells. (**J**) Changes in *Egr3* expression (C9) in hepatic NK cells. Blue, red, and green represent uninfected and four and six week post-infection samples, respectively. * *p* < 0.05, ** *p* < 0.01, *** *p* < 0.001.

**Figure 7 ijms-26-03211-f007:**
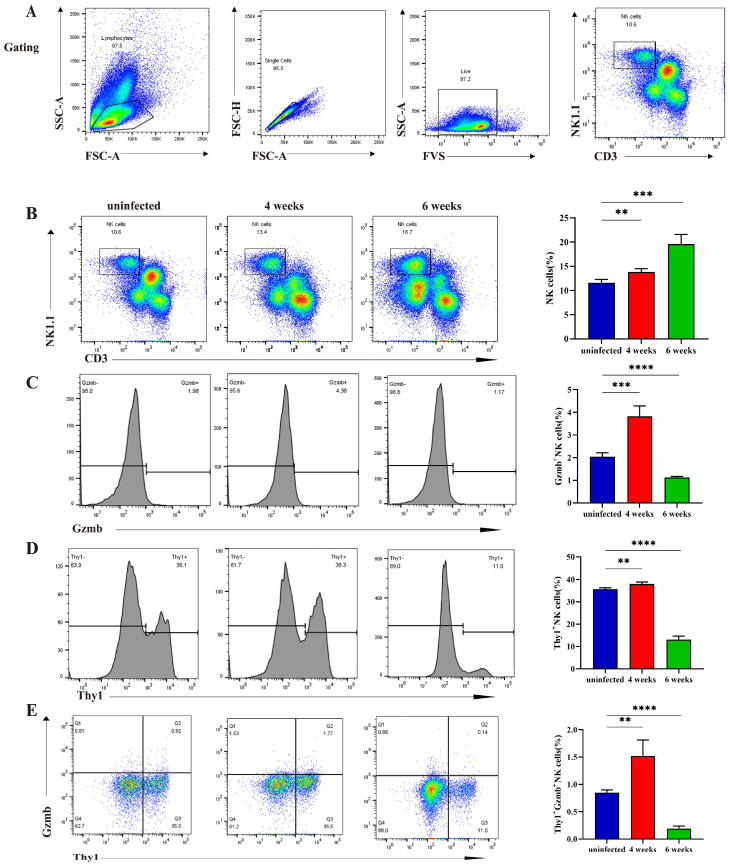
Changes in the proportion and function of hepatic total NK and cluster 4 NK in uninfected mice and mice at four and six weeks post-infection. (**A**) Flow cytometry gating strategy for hepatic lymphocytes, living cells, and NK cells. (**B**) Changes in the proportions of NK cells after infection. (**C**) Changes in Gzmb expression levels of NK cells after infection. (**D**) Changes in Thy1^+^NK cells after infection. (**E**) Changes in Gzmb expression levels of Thy1^+^NK cells after infection. ** *p* < 0.01, *** *p* < 0.001, **** *p* < 0.0001.

**Figure 8 ijms-26-03211-f008:**
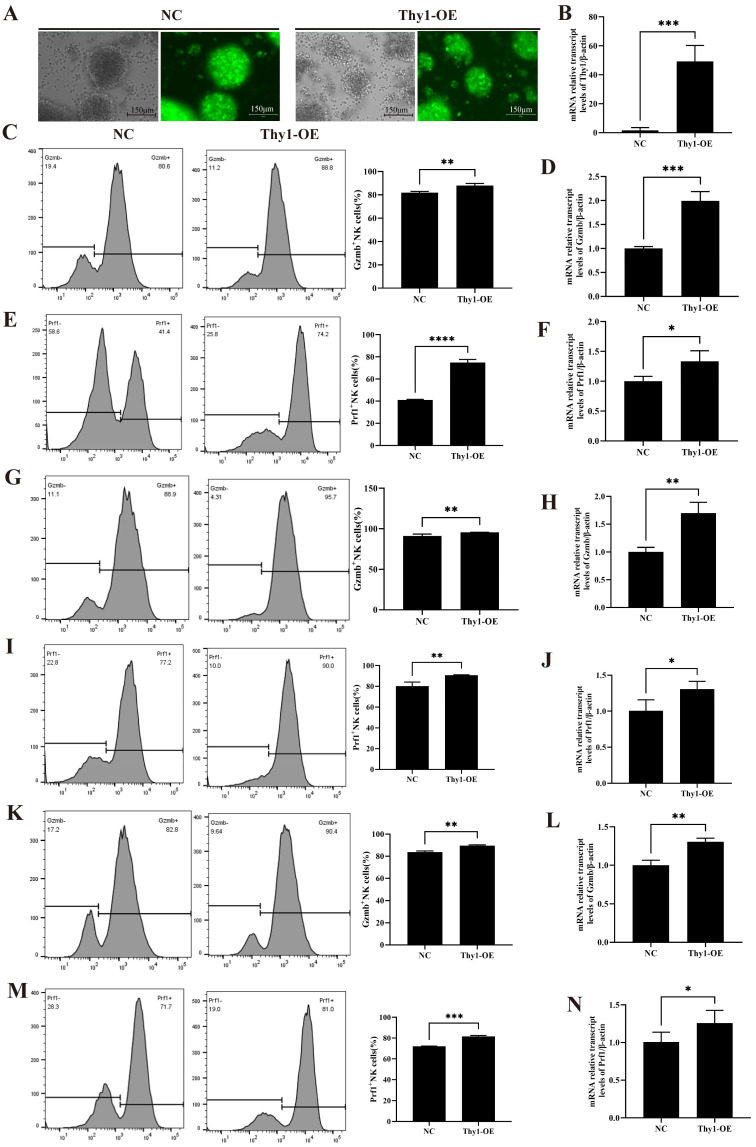
Validation of effector function in Thy1-overexpressing NK cells. (**A**) Green Fluorescent Protein (GFP) expression in Thy1-overexpressing NK92 cells and negative control NK92 cells. (**B**) RT-qPCR validation of the effects of Thy1 overexpression in NK92 cells. (**C**,**D**) Changes in Gzmb expressions of protein and gene levels in NC and Thy1-OE NK92 cells stimulated with IL-12, IL-15, and IL18 for 24 h. (**E**,**F**) Changes in Prf1 expressions of protein and gene levels in NC and Thy1-OE NK92 cells stimulated with IL-12, IL-15, and IL18 for 24 h. (**G**,**H**) Changes in Gzmb expressions of protein and gene levels in NC and Thy1-OE NK92 cells stimulated with IL-12, IL-15, and IL18 for 48 h. (**I**,**J**) Changes in Prf1 expressions of protein and gene levels in NC and Thy1-OE NK92 cells stimulated with IL-12, IL-15, and IL18 for 48 h. (**K**,**L**) Changes in Gzmb expressions of protein and gene levels in NC and Thy1-OE NK92 cells stimulated with SEA for 24 h. (**M**,**N**) Changes in Prf1 expressions of protein and gene levels in NC and Thy1-OE NK cells stimulated with SEA for 24 h. NC: negative control; Thy1-OE: Thy1 overexpression NK cells. * *p* < 0.05, ** *p* < 0.01, *** *p* < 0.001, **** *p* < 0.0001.

**Figure 9 ijms-26-03211-f009:**
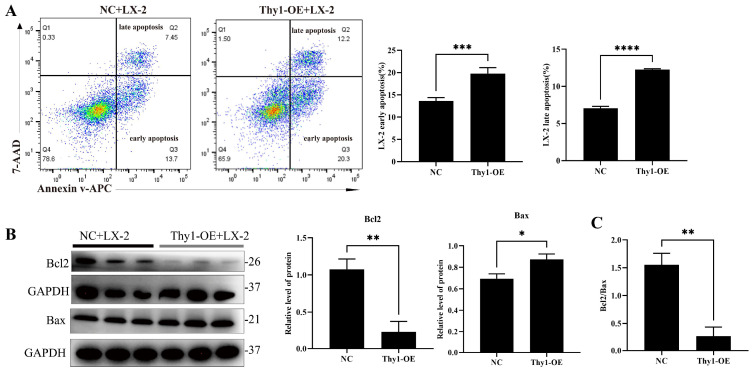
Validation of the cytotoxic function of Thy1-overexpressing NK cells. (**A**) Alterations in the early and late apoptosis ratios of LX-2 cells following 6 h of co-culture with NC and Thy1-OE NK cells. (**B**) Modulations in Bcl2 and Bax protein expression levels in LX-2 cells after 6 h of co-culture with NC and Thy1-OE NK92 cells. (**C**) Changes in the Bcl2: Bax ratio of LX-2 cells. NC: negative control NK92 cells; Thy1-OE: Thy1 overexpression NK92 cells. * *p* < 0.05, ** *p* < 0.01, *** *p* < 0.001, **** *p* < 0.0001.

## Data Availability

Single-cell sequencing data are deposited in the Genome Sequence Archive of the China National Center for Bioinformation with the submission CRA013453 (https://ngdc.cncb.ac.cn/gsa/ (accessed on 31 December 2022)). All data needed to evaluate the conclusions in the paper are present in the paper or the Appendix A.

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
