# Peer review of "Single-Cell Sequencing Reveals the Heterogeneity of Hepatic Natural Killer Cells and Identifies the Cytotoxic Natural Killer Subset in Schistosomiasis Mice"

_ijms, 2025, doi:10.3390/ijms26073211_

Round 1
Reviewer 1 Report
Comments and Suggestions for Authors
In "Single-cell sequencing reveals the heterogeneity of hepatic NK 2 and identifies the cytotoxic NK subset in schistosomiasis mice", Xu et al described the cell type-specific gene expression of hepatic NK cells from mice experimentally infected with Schistosoma japonicum at different times after infection. They provided extensive bioinformatic information of NK cells during Schistosoma infection. The work is technically well done and provides some novel insights in regards to the role of NK cells in Schistosoma infection. There are a few points of clarification that should be addressed.
1. In this study, NK cells were enriched by magnetic beads from liver NPCs. However, there were still other cell types left such as B cell, T cell, etc. These left cells after magnetic bead sorting could not reflect the real cellularity in the liver. Thus, the data description in Supplementary Figure 1 is not accurate. The ratios of four clusters NK cells in three samples should be shown as in Figure 4B.
2. Since our paper focus on NK cells, the other cell types could be shown in supplementary Figures with their canonical markers. In Figure 1C and Figure 2A, the gene expression could focus on the four cluster of NK cells also. Besides, the three dot colors with different expression levels could be illustrated in Figure 1C.
3. Thy1 overexpression could enhance NK cell cytotoxic activity. Authors should comment on this reason.
4. The authors stated that SEA enhanced Gzmb and Prf1 expression levels in Thy1 overexpressed NK cells. Authors should comment on this reason.
5. In Figure 7A, the single cells should be gated in all cells in the diagonal.
Author Response
Comments 1: In this study, NK cells were enriched by magnetic beads from liver NPCs. However, there were still other cell types left such as B cell, T cell, etc. These left cells after magnetic bead sorting could not reflect the real cellularity in the liver. Thus, the data description in Supplementary Figure 1 is not accurate. The ratios of four clusters NK cells in three samples should be shown as in Figure 4B.
Response 1: Thank you for your comments. According to your suggestions, we have made the following modifications to Supplementary Figure 1.
- Supplementary Figure 1A (The number of cells) is deleted.
- In Supplementary Figure 1, a figure (1B named “The ratio of NK cluster cells in total NK in different samples”) is added.
Comments 2: Since our paper focus on NK cells, the other cell types could be shown in supplementary Figures with their canonical markers. In Figure 1C and Figure 2A, the gene expression could focus on the four cluster of NK cells also. Besides, the three dot colors with different expression levels could be illustrated in Figure 1C.
Response 2: Your suggestions are excellent. According to your suggestion, all the clusters in Figure 1C except NK were removed. In Figure 2A, the NK subsets are highlighted by the red frame in the violin plot. In addition, to show Figure 1C more clearly, we also added the following figure notes, “inf_0w_liver_SJ, inf_4w_liver_SJ, inf_6w_liver_SJ” represent uninfected, and four- and six-week post-infection samples, respectively, and the numbers (0, 6, 10, 14) in front of them represent the names of NK clusters. The blue, red, and green dots represent samples that were uninfected, and four- and six-weeks post-infection samples respectively, and the size of the dots represents the level of gene expression.”
Comments 3: Thy1 overexpression could enhance NK cell cytotoxic activity. Authors should comment on this reason.
Response 3: Thanks for your suggestions. We added the following sentences to the discussion section in lines 458-462: “Itami found that Thy1 may bind to integrin β3 [60]. Thy1 inhibition blocked the adhesion of tumor cells to mouse lymphatic endothelial cells [61]. In our study, Thy1 overexpressed NK cells have a strong ability to kill LX-2. It is speculated that thy1 might enhance the adhesion of NK cells to LX-2, thereby killing more LX-2 cells.”
Comments 4: The authors stated that SEA enhanced Gzmb and Prf1 expression levels in Thy1 overexpressed NK cells. Authors should comment on this reason.
Response 4: Thanks for your suggestions. We added the following sentences to the discussion section in lines 462-468: “Kupz observed that Thy1+NK cells could secret protective IFN-γ in the early stages of Salmonella infection, enhancing the host’s antimicrobial immunity [62]. In this study, we used SEA to stimulate NK cells in vitro to mimic the stimulation of NK received after infection in vivo. After stimulation by SEA, Thy1 overexpressed NK cells secreted more Gzmb and Prf1. It indicates that Thy1 can enhance the cytotoxicity of NK cells, but the specific mechanism involved remains unclear and needs further study.”
Comments 5: In Figure 7A, the single cells should be gated in all cells in the diagonal.
Response 5: Thank you for your suggestion. In previous flow cytometry experiments, we used CD3-NK1.1+ to perform reverse gating on non-parenchymal cells (NPCs) to define the origin of NK cells and found NK belongs to lymphocytes. Therefore, in this study, we performed gating on lymphocytes, excluding other cell types, and then used CD3-NK1.1+ to define NK cells accurately.
Reviewer 2 Report
Comments and Suggestions for Authors
In this study, the authors isolated NK cells from S. japonicum infected mice and compared with NK cells from NC mice by single cell RNA-sequencing. The results showed that hepatic NK cells could be divided into mature, immature, regulatory-like, and memory-like NK cells and re-clustered into ten subsets. Two subsets (C3 and C4) increased after S. japonicum infection which might be associated with the resistance of S. japonicum infection. The authors also validated that NK cells overexpressing Thy1, a marker gene in C4, enhanced the HSC apoptosis than the control NK cells, therefore Thy1+NK cells might be potential target cells against liver fibrosis.
Overall, the authors have done a substantial amount of work. The design of the experiments is reasonable and the results seem to support their conclusion. The novelty of this study is good because this seems to be the first study going deep into the different subsets of hepatic NK cells and their potential roles in the pathogenesis of S. japonicum induced liver fibrosis, which is quite interesting to me. However, the writing of the manuscript and the presenting of the data can be further improved. I would suggest a minor revision. The authors should revise and improve their manuscript according to the concerns/question listed below.
1. Line 101-106. The NK cells were categorized into mature, immature, memory-like, and regulatory-like NK cells. What are the criteria of such categorization? It is not clearly explained in the methods section. What are the marker genes used for cell type identification? I don’t find relevant information in the manuscript (correct me If I’m wrong). Please give enough details of your cell type categorization.
2. Similarly, in Line 85-86, What are the marker genes used to define NK cells, NKT cells, neutrophils, monocyte–macrophages, and MDSCs ? It is not mentioned in Fig.1B, nor could I find it elsewhere in the manuscript and supplementary.
3. Line 442. Why is each sample a mixture of NK cells from 6 mice instead of 6 samples from 6 mice as biological replicates? Is it because not enough NK cells can be isolated from one mouse? Or because it is too expensive to perform 6 single cell RNA sequencing in each group?
4. Some expression are ambiguous. e.g. Line 88-90, gene expression(s), expression of what genes? I’m confused. In Fig.1C, I think the “0_0w”, “0_4w”, and “0_6w” at the bottom of the figure represent the three samples of C0 NK cells. The solid/fade dots represent the gene expression levels. But I don’t see a obvious difference in week 0, week 4 and week 6. Can you clarify the genes that changed significantly? Furthermore, What do the green, red, blue color mean? I guess they represent week 6, week 4, and week0 samples, respectively. However, it is not clearly indicated in the figure legend, so a lot of confusion is made.
5. Similar to Fig. 1C, other figures, in my personal opinion, are also not very clearly explained in figure legends. I understand most figures are typical figures used in single cell RNA sequencing, and I saw figures presented in the same way in literature, so I understand such way of presentation is OK for readers familiar with single cell RNA sequencing. However, I still personally suggest more details and explanation (e.g what does each mark in the figure mean) should be included in the figures/figure legend. Otherwise, it would be quite difficult for readers who have not come across single cell RNA sequencing to understand.
6. The results can be refined. In my personal opinion, A common issue in high-throughput omics articles is that their results sections often resemble a long shopping list, making them very monotonous and uninteresting, leaving readers unsure about the main points. In this articles, I also see a long list of genes in each subset. But what do the high expression of these genes mean, and what their biological significance in the context of S. japonicum infection are, are not well explained in the results section, nor in the discussion. Furthermore, the discussion section is somewhat like a repeat of the result section, but the interpretation of your data, and how the results helps to further understand the mechanism of NK cells inhibits liver fibrosis, is not well discussed. For example, after reading the pseudo time analysis of NK subsets, I understand the details of the results, but the purpose of such analysis is not clearly explained. What is your conclusion in this section? Can you summarize your finding in each section in one or two sentences?
7. Line 298-311. There are many markers genes in the 10 NK subgroups. So why is only Thy1 chosen and validated in NK92 cells overexpressed Thy1 is not clearly explain. I guess it is because Thy1 is the marker gene for C4 subset and increased significantly 4 weeks after infection. However it is suggested to explicitly explain the reason to chose and validate Thy1. Similarly, why Gzmb and Prf1 expression were measured should also be explained.
8. Some minor errors in English. e.g. Line 167, 170, and so on, S. japonicum should be “S. japonicum”.
Author Response
Comments 1: Line 101-106. The NK cells were categorized into mature, immature, memory-like, and regulatory-like NK cells. What are the criteria of such categorization? It is not clearly explained in the methods section. What are the marker genes used for cell type identification? I don’t find relevant information in the manuscript (correct me If I’m wrong). Please give enough details of your cell type categorization.
Response 1: Thanks for your good suggestion. According to marker expression, GO, and KEGG analyses, four NK groups were identified as mature NK, immature NK, memory-like NK, and regulatory-like NK cells. We revised these contents in discussion from lines 402 to 442.
Comments 2: Similarly, in Line 85-86, What are the marker genes used to define NK cells, NKT cells, neutrophils, monocyte–macrophages, and MDSCs ? It is not mentioned in Fig.1B, nor could I find it elsewhere in the manuscript and supplementary.
Response 2: Thank you for your good suggestion. The markers that define the cell type have been added in lines 86-87 as follows: “these cells could be defined as NK cells (NK1.1+NKp46+), NKT cells (CD3+NK1.1+), neutrophils (Ly6G+CXCR2+), monocyte-macrophages (CD14+), and MDSCs (CD11b+Ly6G+) (Figure 1B).”
Comments 3: Line 442. Why is each sample a mixture of NK cells from 6 mice instead of 6 samples from 6 mice as biological replicates? Is it because not enough NK cells can be isolated from one mouse? Or because it is too expensive to perform 6 single cell RNA sequencing in each group?
Response 3: Thank you for your good suggestion. It is challenging to isolate enough NK cells from a single mouse, and it is too expensive to sequence single-cell RNA from six samples per group.
Comments 4: Some expression are ambiguous. e.g. Line 88-90, gene expression(s), expression of what genes? I’m confused. In Fig.1C, I think the “0_0w”, “0_4w”, and “0_6w” at the bottom of the figure represent the three samples of C0 NK cells. The solid/fade dots represent the gene expression levels. But I don’t see a obvious difference in week 0, week 4 and week 6. Can you clarify the genes that changed significantly? Furthermore, What do the green, red, blue color mean? I guess they represent week 6, week 4, and week0 samples, respectively. However, it is not clearly indicated in the figure legend, so a lot of confusion is made.
Response 4: Thank you for your good suggestion. The description of Fig. 1C in lines 90-93 has been revised as: “Klra7 had the highest gene expression level at the fourth-week post-infection, and the expression level at the sixth-week was lower the fourth-week post-infection (Figure 1C). The expression levels of other genes showed similar trends.” In addition, to show Fig. 1C more clearly, we also added the following figure notes: “inf_0w_liver_SJ, inf_4w_liver_SJ, inf_6w_liver_SJ” represent uninfected, and four- and six-week post-infection samples, respectively, and the numbers (0, 6, 10, 14) in front of them represent the names of NK clusters. The blue, red, and green dots represent samples that were uninfected, and four- and six-weeks post-infection samples respectively, and the size of the dots represents the level of gene expression.”
Comments 5: Similar to Fig. 1C, other figures, in my personal opinion, are also not very clearly explained in figure legends. I understand most figures are typical figures used in single cell RNA sequencing, and I saw figures presented in the same way in literature, so I understand such way of presentation is OK for readers familiar with single cell RNA sequencing. However, I still personally suggest more details and explanation (e.g what does each mark in the figure mean) should be included in the figures/figure legend. Otherwise, it would be quite difficult for readers who have not come across single cell RNA sequencing to understand.
Response 5: Your suggestions are excellent. Per your suggestion, the following figure notes have been added to the main text. The parts highlighted in yellow are modified.
(1) Fig. 1: inf_0w_liver_SJ, inf_4w_liver_SJ, inf_6w_liver_SJ” represent uninfected, and four- and six-week post-infection samples, respectively, and the numbers (0, 6, 10, 14) in front of them represent the names of NK clusters. The blue, red, and green dots represent samples that were uninfected, and four- and six-weeks post-infection samples respectively, and the size of the dots represents the level of gene expression.
(2) Fig. 2: (A) Violin plots of the top three markers in the four NK cell clusters. According to the order circled in red, from left to right, there are four clusters of NK: cluster 0, cluster 6, cluster 10, and cluster 14. (B) GO enrichment analysis of the four NK cell clusters. (C) KEGG enrichment analysis of the four NK cell clusters. The size of the point represents the number of differentially expressed genes of the term; the color of the points represents the value of FDR. (D) Comparison of significant pathways among the four NK clusters. The height of the bar graph represents the pathway expression level.
(3) Fig. 3: The red indicates that the pathway expression level is higher in this cluster, while the blue indicates that the pathway expression level is lower in this cluster.
(4) Fig. 4: (A) Violin plots of the top then markers in cluster 3 and cluster 4. (B) GO enrichment analysis of cluster 3 and cluster 4. (C) KEGG enrichment analysis of cluster 3 and cluster 4. The size of the point represents the number of differentially expressed genes of the term; the color of the points represents the value of FDR. (D) Comparison of significant pathways among the ten subsets. The height of the bar graph represents the pathway expression level. (E) Analysis of six aspects of function correlated with signal pathways among the four clusters and ten subsets of NK cells. The size of the circle represents the number of signal pathways. The color of the circle corresponds to different cellular functions.
(5) Fig. 5: (A) Monocle pseudo-time of hepatic NK cells. 1, 2, and 3 in the circle represent branch points 1, 2, and 3, respectively.
(6) Fig. 6: (J) Changes in Egr3 expression (C9) in hepatic NK cells. Blue, red, and green represent uninfected and four- and six-week post-infection samples, respectively.
Comments 6: The results can be refined. In my personal opinion, A common issue in high-throughput omics articles is that their results sections often resemble a long shopping list, making them very monotonous and uninteresting, leaving readers unsure about the main points. In this articles, I also see a long list of genes in each subset. But what do the high expression of these genes mean, and what their biological significance in the context of S. japonicum infection are, are not well explained in the results section, nor in the discussion. Furthermore, the discussion section is somewhat like a repeat of the result section, but the interpretation of your data, and how the results helps to further understand the mechanism of NK cells inhibits liver fibrosis, is not well discussed. For example, after reading the pseudo time analysis of NK subsets, I understand the details of the results, but the purpose of such analysis is not clearly explained. What is your conclusion in this section? Can you summarize your finding in each section in one or two sentences?
Response 6: Thank you for your good suggestion. According to your suggestion, we have made some revisions to present our options.
2.2: In lines 134-137, “According to the gene expressions and enrichment analysis using the Kyoto Encyclopedia of Genes and Genomes (KEGG) and Gene Ontology (GO) databases, the hepatic NK cells could be categorized into four types, including mature NK (cluster 0), immature NK (cluster 6), memory-like NK (cluster 10), and regulatory-like NK (cluster 14) cells.”
2.3: In lines 163-167, “The GSVA analysis indicates that memory-like NK cells exhibited strong inflammatory responses and weak proliferative characteristics, regulatory-like NK cells highly expressed immune-inhibitory pathways, mature cells mainly activated reactome lectin pathway, complement activation, and integrin pathway, while immature cells significantly activated proliferation and differentiation-related pathways.”
2.4: In lines 221-224, “In this part, the most interesting thing is the proportion of C3 and C4 changes increased significantly with the progression of infection. They may play important roles in the schistosomiasis liver fibrosis. C3 is similar to CD8+ T cells involved in antigen presentation. The proteasome-related pathway in C4 is the most activity.”
2.5: In lines 273-276, “The pseudo-time analysis reveals that persistent infection suppresses NK function. The immature NK cells are at the starting point of differentiation. The mature NK cells and regulatory-like NK cells are located in the middle of the differentiation trajectory. The memory-like NK cells are terminally differentiated NK, which have a complex function.”
Comments 7: Line 298-311. There are many markers genes in the 10 NK clusters. So why is only Thy1 chosen and validated in NK92 cells overexpressed Thy1 is not clearly explain. I guess it is because Thy1 is the marker gene for C4 subset and increased significantly 4 weeks after infection. However it is suggested to explicitly explain the reason to chose and validate Thy1. Similarly, why Gzmb and Prf1 expression were measured should also be explained.
Response 7: The reasons for choosing Thy1 have been added in lines 309-311 as follows:
“The expression level of thy1 in the C4 subset is the highest, and it can be regarded as the surface marker of the C4 cluster.” The reason for detecting the expression of Gzmb and Prf1 is as follows: Granzyme B (Gzmb) and perforin (Prf1) are important effector molecules of the cytotoxic function of NK cell. Perforin forms a transmembrane channel on the surface of target cells and increases membrane permeability, and granzyme enters the interior of target cells and induces apoptosis of target cells. We detected the expression of Gzmb and Prf1 to measure the NK cytotoxic function.
Comments 8: Some minor errors in English. e.g. Line 167, 170, and so on, S. japonicum should be “S. japonicum”.
Response 8: Thanks for your suggestion. "S. japonicum" has been replaced with "S. japonicum".
Reviewer 3 Report
Comments and Suggestions for Authors
Liver tissue fibrosis is a complex process that has not yet been fully understood. The current state of knowledge allows us to conclude that not only liver cells but the whole microenvironment, including also immune system cells or epigenetic changes are involved in this process.
The submitted work for review addresses this knowledge gap, offering novel insights into the fibrosis process and the role of NK cells.
The manuscript's introduction methodically familiarises the reader with the subject matter, while the research hypothesis lucidly encapsulates the scope and significance of the research undertaken.
However, the quality and presentation of the results could be improved, particularly with regard to the dimensions of the figures and their accompanying legends. At present, the description of the figures is inadequate. It is recommended that details such as comparisons to the author's own studies (line 93) or references to other studies be relocated to the discussion section. Additionally, the description of results should be expanded to incorporate information that is crucial for the reader. For instance, the term "highly expressed" requires clarification, as it is not clear whether this refers to a fold change of 1.5 or more precisely 30 (see lines 107-108, 119-120, etc.). Similarly, the reference to "many genes being highly expressed" (line 125) is imprecise and requires rephrasing.
Furthermore, the current form of Figures 2-5 is deemed unacceptable and consideration should be given to a different form of presentation of these results (or enlarging these figures). The descriptions under the figures (Figures 2-6) are laconic, without a legend and a precise presentation of what is seen in the figure and why it is important enough to show it in the publication.
It is evident that the information contained within lines 300-301 pertains to the methodological framework. To ensure clarity and conciseness, it would be advisable to either modify the presentation of this information or to relocate it to the section designated for the description of methods.
The research methodology and bioinformatic analyses employed are both correct and consistent with the research issues. An interesting complement to this type of research would be to conduct spatial transcriptomics analysis or XENIUM analysis for these trials, with a view to exploring the role of NK cells in fibrosis processes.
The discussion is written in an unorganised manner, leaving the reader with many unanswered questions and lacking the classic division into references to the work of other researchers that are consistent with and in opposition to the obtained results. There is also a lack of a critical view of the authors on their own work, which should be presented in the "limitation" section.
With regard to the research questions that emerged during the review of this article:
- What is the translational potential of the research conducted? The authors employed co-cultures of NK cells and LX-2, but the origin of the NK cells remains ambiguous, as the text fails to clarify whether they were human or mouse. Liver fibrosis is a multifaceted process, and concerns have been raised that the observations made in mice may not have translational or clinical significance.
- Secondly, it is important to consider whether NK cells are the only cells of the immune system that play a role in liver tissue, and whether other populations of cells (e.g. macrophages) interact with them. The lack of analyses and explanations in this direction makes it difficult to fully understand the multidirectional interaction in the process of liver fibrosis.
- Thirdly, the research model itself remains controversial. While parasite-induced fibrosis has some impact on the clinic, autoimmune-induced fibrosis or even alcohol consumption could be a more appropriate research model.
- The next point pertains to the question of whether the authors have conducted research on modified natural killer (NK) cells that have been transferred to infected mice.
- Secondly, while the number of samples used in the experiment is small, it is believed that this number is sufficient to conduct the necessary bioinformatic analyses due to the costs of the method used.
In conclusion, the article provides valuable insights into the process of liver fibrosis. However, it would be advisable to subject it to substantial editing and consider altering its length. It may be beneficial to shorten the article and present the results in a different format. The focus should be directed towards the scRNA analyses themselves, and the article should be expanded to provide a justification for the selected research model and the conclusions derived from it.
Author Response
Comments 1: However, the quality and presentation of the results could be improved, particularly with regard to the dimensions of the figures and their accompanying legends. At present, the description of the figures is inadequate. It is recommended that details such as comparisons to the author's own studies (line 93) or references to other studies be relocated to the discussion section. Additionally, the description of results should be expanded to incorporate information that is crucial for the reader. For instance, the term "highly expressed" requires clarification, as it is not clear whether this refers to a fold change of 1.5 or more precisely 30 (see lines 107-108, 119-120, etc.). Similarly, the reference to "many genes being highly expressed" (line 125) is imprecise and requires rephrasing.
Response 1: Thanks for your suggestion. The single-cell sequencing analysis of this study involves a large number of related figures, so each figure is quite large. We modified the figures as follows:
(1) In Figure 1C, only the NK clusters were retained, while deleted the other non-NK clusters. (2) In Figure 2A, we changed the violin plot showing the top 10 genes to the top 3. The distribution of the small figures within Figure 2 has been adjusted. (3) We moved the original Figures 4A and 4B to Supplementary Figure 2. The distribution of the small figures within Figure 4 has been adjusted. (4) We moved Figure 5F to Supplementary Figure 6.
According to your suggestions, the sentence on line 93, "This trend aligned with our earlier findings on NK cell function [25]. " has been moved to the discussion section in lines 388-389. To facilitate readers' more profound understanding of the content of this study, we have added a summary at the end of each part of the results to present our essential results. The fold changes of gene expression for the "high expression" genes in each subgroup are not quite the same, and they are generally greater than 1.5.
Comments 2: Furthermore, the current form of Figures 2-5 is deemed unacceptable and consideration should be given to a different form of presentation of these results (or enlarging these figures). The descriptions under the figures (Figures 2-6) are laconic, without a legend and a precise presentation of what is seen in the figure and why it is important enough to show it in the publication.
Response 2: Thanks for your suggestion. According to your suggestion, we modified the figures as described in the response to Q1.
To facilitate readers' understanding of the contents expressed in each figure, we have added textual explanations below in Figures 1 to 6. The specific content is as follows:
(1) Fig. 1: inf_0w_liver_SJ, inf_4w_liver_SJ, inf_6w_liver_SJ” represent uninfected, and four- and six-week post-infection samples, respectively, and the numbers (0, 6, 10, 14) in front of them represent the names of NK clusters. The blue, red, and green dots represent samples that were uninfected, and four- and six-weeks post-infection samples respectively, and the size of the dots represents the level of gene expression.
(2) Fig. 2: (A) Violin plots of the top three markers in the four NK cell clusters. According to the order circled in red, from left to right, there are four clusters of NK: cluster 0, cluster 6, cluster 10, and cluster 14. (B) GO enrichment analysis of the four NK cell clusters. (C) KEGG enrichment analysis of the four NK cell clusters. The size of the point represents the number of differentially expressed genes of the term; the color of the points represents the value of FDR. (D) Comparison of significant pathways among the four NK clusters. The height of the bar graph represents the pathway expression level.
(3) Fig. 3: The red indicates that the pathway expression level is higher in this cluster, while the blue indicates that the pathway expression level is lower in this cluster.
(4) Fig. 4: (A) Violin plots of the top then markers in cluster 3 and cluster 4. (B) GO enrichment analysis of cluster 3 and cluster 4. (C) KEGG enrichment analysis of cluster 3 and cluster 4. The size of the point represents the number of differentially expressed genes of the term; the color of the points represents the value of FDR. (D) Comparison of significant pathways among the ten subsets. The height of the bar graph represents the pathway expression level. (E) Analysis of six aspects of function correlated with signal pathways among the four clusters and ten subsets of NK cells. The size of the circle represents the number of signal pathways. The color of the circle corresponds to different cellular functions.
(5) Fig. 5: (A) Monocle pseudo-time of hepatic NK cells. 1, 2, and 3 in the circle represent branch points 1, 2, and 3, respectively.
(6) Fig. 6: (J) Changes in Egr3 expression (C9) in hepatic NK cells. Blue, red, and green represent uninfected and four- and six-week post-infection samples, respectively.
Comments 3: It is evident that the information contained within lines 300-301 pertains to the methodological framework. To ensure clarity and conciseness, it would be advisable to either modify the presentation of this information or to relocate it to the section designated for the description of methods.
Response 3: Your suggestion is excellent. The sentences have been revised in lines 327-328: “Figure 8A shows that both groups of cells exhibit green fluorescence, indicating that the lentivirus has been successfully transfected into the cells.”
Comments 4: The research methodology and bioinformatic analyses employed are both correct and consistent with the research issues. An interesting complement to this type of research would be to conduct spatial transcriptomics analysis or XENIUM analysis for these trials, with a view to exploring the role of NK cells in fibrosis processes.
Response 4: Your suggestion is extremely valuable. We also plan to incorporate more spatial transcriptomics analysis or XENIUM analysis in our future research to explore the role of NK cells in the fibrosis process.
Comments 5: The discussion is written in an unorganised manner, leaving the reader with many unanswered questions and lacking the classic division into references to the work of other researchers that are consistent with and in opposition to the obtained results. There is also a lack of a critical view of the authors on their own work, which should be presented in the "limitation" section.
Response 5: Thank you for your excellent suggestions. In lines 369-376, we added comments on the influencing factors of liver fibrosis. In lines 402-442, we revised the comments on the functions and differentiation trajectories of various NK subsets such as mature NK cells, immature NK cells, memory-like NK cells, regulatory-like NK cells, and NK cells with C0-C9. In lines 458-468, we added comments on the mechanism by which Thy1 enhances the cytotoxicity of NK cells. Besides this, we have added a discussion on the limitations of this study in the last paragraph of this article. In lines 473-477, we said as follows: “However, the mechanism by which Thy1 enhances the cytotoxicity of NK cells is unclear. This study provides clues for further searching for NK cell clusters with high cytotoxicity. However, the limitation is we don’t explore Thy1+NK’ role in liver fibrosis in vivo. In the future, we will continue to verify Thy1+NK’ role against liver fibrosis in animals.”
Comments 6: What is the translational potential of the research conducted? The authors employed co-cultures of NK cells and LX-2, but the origin of the NK cells remains ambiguous, as the text fails to clarify whether they were human or mouse. Liver fibrosis is a multifaceted process, and concerns have been raised that the observations made in mice may not have translational or clinical significance.
Response 6: Thank you for your suggestion. The NK92 and LX-2 cells used in this study are human-derived cell lines. “human cell line” has been added to lines 558 and 568.
Comments 7: Secondly, it is important to consider whether NK cells are the only cells of the immune system that play a role in liver tissue, and whether other populations of cells (e.g. macrophages) interact with them. The lack of analyses and explanations in this direction makes it difficult to fully understand the multidirectional interaction in the process of liver fibrosis.
Response 7:Thank you for your good comment. Based on your suggestion, the following sentences have been added to the discussion section in lines 369-376: “The development of liver fibrosis is regulated by various factors including immune cells, cytokines, etc.[25]. M2 macrophages induce hepatic stellate cells to undergo autophagy through the PGE2/EP4 pathway to promote liver fibrosis in NAFLD mice [26]. Macrophages are induced to undergo M1-type polarization by MyD88 in hepatic stellate cells, which can enhance liver fibrosis [27]. CD8+ tissue-resident memory T cells inhibit hepatic fibrosis by inducing hepatic stellate cell apoptosis [28]. Natural killer cells (NK) are important immune cells in the liver. In recent years, the anti-fibrotic effect of NK has attracted much attention from researchers.”
Comments 8: Thirdly, the research model itself remains controversial. While parasite-induced fibrosis has some impact on the clinic, autoimmune-induced fibrosis or even alcohol consumption could be a more appropriate research model.
Response 8: Thank you for your good suggestion. Our research group has been studying S. japonicum for decades and has a mature system for constructing a schistosomiasis liver fibrosis model in mice. Therefore, we used this model in our study. In the future, we will explore other models of liver fibrosis, such as the carbon tetrachloride model, alcoholic liver fibrosis model, autoimmune-induced fibrosis, etc.
Comments 9: The next point pertains to the question of whether the authors have conducted research on modified natural killer (NK) cells that have been transferred to infected mice.
Response 9: Thank you very much for your suggestion. We have already constructed mice with overexpressed Thy1. Next, we will isolate NK cells from these mice and transfer the isolated cells back into the infected mice with S. japonicum to observe the effect on liver fibrosis. This will help us verify the anti-fibrotic role of Thy1+NK cells.
Comments 10: Secondly, while the number of samples used in the experiment is small, it is believed that this number is sufficient to conduct the necessary bioinformatic analyses due to the costs of the method used.
Response 10: I appreciate your understanding.
Comments 11: In conclusion, the article provides valuable insights into the process of liver fibrosis. However, it would be advisable to subject it to substantial editing and consider altering its length. It may be beneficial to shorten the article and present the results in a different format. The focus should be directed towards the scRNA analyses themselves, and the article should be expanded to provide a justification for the selected research model and the conclusions derived from it.
Response 11:Thanks for your good suggestion. These comments are all valuable and very helpful for improving our paper. We have studied the comments carefully and have revised our manuscript. We hope the revised manuscript will meet your requirements.
Round 2
Reviewer 1 Report
Comments and Suggestions for Authors
No comments.
Reviewer 3 Report
Comments and Suggestions for Authors
The manuscript has not undergone a sufficient level of revision and, when compared to the previous version, falls well below the required standard. Furthermore, the explanations provided by the authors are inadequate. Given the significant deficiencies identified, the paper does not meet the necessary criteria for publication.It is strongly recommended that a thorough revision be undertaken, including the addition of further analyses, before the manuscript is submitted to a journal that better aligns with the research scope and current developments in the field. However, substantial improvements to the methodology, structure, and depth of the discussion are essential before pursuing publication elsewhere.